# Effect of Task Constraints on Neurobiological Systems Involved in Postural Control in Individuals with and without Chronic Ankle Instability

**DOI:** 10.3390/bioengineering11100956

**Published:** 2024-09-25

**Authors:** Yuki A. Sugimoto, Patrick O. McKeon, Christopher K. Rhea, Carl G. Mattacola, Scott E. Ross

**Affiliations:** 1Department of Physical Therapy & Human Movement Science, Feinberg School of Medicine, Northwestern University, Chicago, IL 60611, USA; 2Department of Exercise Science & Athletic Training, Ithaca College, Ithaca, NY 14850, USA; pmckeon@ithaca.edu; 3Ellmer College of Health Sciences, Old Dominion University, Norfolk, VA 23529, USA; crhea@odu.edu; 4The School of Health & Human Sciences, The University of North Carolina at Greensboro, Greensboro, NC 27412, USA; cgmattac@uncg.edu (C.G.M.); seross@uncg.edu (S.E.R.)

**Keywords:** nonlinear dynamics, sensory reweighting system, postural control, ankle sprains, task constraints

## Abstract

The purpose of this study is to investigate the effect of task constraints on the neurobiological systems while maintaining postural control under various sensory feedback manipulations in individuals with and without Chronic Ankle Instability (CAI). Forty-two physically active individuals, with and without CAI, were enrolled in a case-control study conducted at a biomechanics research laboratory. All participants underwent the Sensory Organization Test (SOT), which assesses individuals’ ability to integrate somatosensory, visual, and vestibular feedback to maintain postural control in double-, uninjured-, and injured-limb stances under six different conditions in which variations in the sway-referenced support surface (platform) and visual surroundings, with and without vision, are manipulated to affect somatosensory and visual feedback. Center-of-Pressure (COP) path length was computed from raw data collected during trials of each SOT condition. Sample Entropy (SampEN) values were extracted from the COP path length time series to examine neurobiological systems complexity, with lower SampEN values indicating more predictable and periodic (rigid) neurobiological systems, while higher SampEN values indicate more unpredictable and random systems. The results show that specific task constraints affect the neurobiological systems. Specifically, individuals with CAI demonstrated reduced complexity (decreased SampEN values) in the neurobiological systems during the uninjured-limb stance when all sensory feedback was intact and during both uninjured- and injured-limb stances when they were forced to rely on vestibular feedback. These results highlight the interplay between sensory feedback and task constraints in individuals with CAI and suggest potential adaptations in the neurobiological systems involved in postural control.

## 1. Introduction

Chronic Ankle Instability (CAI) is a significant clinical concern due to its prevalence and the substantial impact on individuals’ quality of life and the healthcare system. Lateral ankle sprains are the most common injury associated with physical activity and sports, accounting for many emergency department visits each year [1]. Despite the perception that ankle sprains are minor injuries that will heal on their own, more than 40% of individuals who experience an ankle sprain develop CAI [1]. This condition is characterized by recurrent ankle sprains and a persistent sensation of the ankle “giving way”, resulting in permanent residual symptoms such as somatosensory dysfunction and impaired postural control. These deficits can lead to complications such as post-traumatic osteoarthritis.

Maintaining postural control is essential for performing daily tasks and preventing joint segment displacement. Successful maintenance of the body’s center of mass within the base of support relies on relevant sensory inputs from the somatosensory, visual, and vestibular systems to coordinate neuromuscular control and thereby, provide dynamic ankle stability. As tasks become more challenging, supraspinal motor control enhances movement precision by minimizing and correcting oscillations at the ankle [2,3]. Individuals with CAI exhibit postural instability and decreased H-reflex modulation in the soleus and peroneous longus muscles during the transition from a bipedal to an unipedal stance [2,3]. However, most postural control studies have focused on the Center-of-Pressure (COP) area, velocity, and excursion length and have not fully estimated the complexity of the neurobiological systems in response to changes in task difficulty (e.g., bipedal to unipedal) [4].

There are nearly infinite ways to maintain postural control due to the high-dimensional degree-of-freedom (DOF) of the human body, which includes independent yet functionally redundant elements from the microscopic levels (e.g., sensory feedback) to the macroscopic levels (e.g., joints) [5,6]. For example, the sensory redundancy of the ankle, knee, and hip exceeds the minimum requirements for postural control. Because the central nervous system (CNS) cannot continuously manage the myriad configurations available, the sensory reweighting system distributes weight to the most relevant sensory feedback specific to maintaining postural control, interacting with organismic (individuals’ health status), task, and environmental constraints [7,8]. In our previous study, we found that the sensory reweighting system was dependent on task constraints [8]. When the task became challenging, both healthy individuals and individuals with CAI relied on vision to maintain posture in the injured-limb stance [8]. However, individuals with CAI relied more on vestibular feedback and showed better postural control in the injured-limb stance compared to healthy individuals [8].

Current evidence suggests that individuals with CAI have a less flexible and adaptive sensorimotor system [9]. As a result, the use of entropy metrics such as Sample Entropy (SampEN) provides a better estimate and quantifies the complexity of the neurobiological systems in individuals with CAI compared to healthy controls [10]. Greater complexity in the neurobiological systems results in optimal postural control, whereas lower complexity leads to either less or more predictable postural control [11]. However, it is still unclear how the complexity of the neurobiological systems changes as the task becomes more difficult while maintaining postural control in different environments designed to manipulate somatosensory and visual feedback, including a combination of a sway-referenced support surface (platform) and visual surroundings with and without vision.

The purpose of this study is to investigate the effect of task constraints on the neurobiological systems while maintaining postural control under various sensory feedback manipulations in individuals with and without CAI. We hypothesized that individuals with CAI will demonstrate significantly lower complexity in the neurobiological systems as task constraints increase, particularly under greater sensory feedback manipulations, while maintaining posture in the injured-limb stance compared to double- and uninjured-limb stances.

## 2. Methods

### 2.1. Study Design

We conducted a case-control study to examine how task constraints affect the neurobiological systems involved in postural control in individuals with and without CAI in a biomechanics research laboratory.

### 2.2. Participants

Forty-two physically active participants, including individuals with unilateral CAI and healthy controls, were recruited (Table 1). CAI was classified according to the International Ankle Consortium criteria [12]. Healthy controls were matched to CAI participants based on sex, age (within ±2 years), height (within ±5%), mass (within ±3%), limb dominance (defined as the leg used to kick a ball), and National Aeronautics and Space Administration Physical Activity Status Scale [NASA-PASS] score (within ±1 on the scale) to assign an injured limb. If a CAI participant’s affected limb was dominant, the corresponding limb for the control participant was also dominant, and vice versa. All participants provided informed consent, and the study protocol was approved by the Institutional Review Board at the University of North Carolina at Greensboro.

### 2.3. Procedure

All participants completed a single session in a biomechanics research laboratory. Upon arrival, they completed a standardized medical history questionnaire, covering previous lower extremity injuries, rehabilitation after ankle sprains, self-reported ankle instability and function, and physical activity status. After a 5-min warm-up on a self-selected intensity bike, they underwent assessments including demographic data (height and weight); joint hypermobility tests; lower extremity anatomical alignment measurements; and postural control tests in double-, uninjured-, and injured-limb stances.

This study was part of the larger research study, and the hypermobility assessments, anatomical tests, and certain conditions of the postural control assessments for double-, uninjured-, and injured-limb stances are not reported in the current study. Participants stood barefoot on a NeuroCom dynamic posturography platform (SMART EquiTest, NeuroCom International Inc. Clackmas, OR, USA) for double, uninjured, and injured limbs with a vest and safety harness, as previously described by Sugimoto et al. [8].

#### 2.3.1. Sensory Organization Test (SOT)

The SOT is a gold-standard assessment of individuals’ ability to integrate somatosensory, visual, and vestibular feedback into postural control. The SOT consists of six conditions (Table 2) in which a combination of a sway-referenced support surface (platform) and visual surroundings, with and without vision, is manipulated to influence somatosensory and visual feedback. These conditions progressively escalate in complexity, isolating different sensory systems, and are designed as follows: 1-Normal Vision-Fixed Surroundings-Fixed Platform (C1-V_n_S_f_P_f_); 2-Absent Vision-Fixed Surroundings-Fixed Platform (C2-V_a_S_f_P_f_); 3-Distorted Vision-Moving Surroundings-Fixed Platform (C3-V_d_S_m_P_f_); 4-Distorted Vision-Fixed Surroundings-Moving Platform (C4-V_d_S_f_P_m_); 5-Absent Vision-Fixed Surroundings-Moving Platform (C5-V_a_S_f_P_m_); and 6-Distorted Vision-Moving Surroundings-Moving Platform (C6-V_d_S_m_P_m_).

During the single-limb stance, participants aligned the medial malleoli of the tested ankle perpendicular to the transverse axis of platform rotation, with the foot positioned centrally. They were instructed to maintain a forward-facing posture, keep their arms relaxed at their sides, and minimize movement while performing the SOT. Each condition consisted of three 20-second trials, for a total of 18 trials per stance. Participants had a 30-second rest between trials and a 1-min rest between conditions. The SOT is traditionally designed for a bipedal stance, making it extremely difficult to maintain posture in a unilateral stance with a sway-referenced support surface, especially for individuals with CAI. To complete the 20-second trials on the injured limb, participants were allowed to tap down with the non-stance toe after 10 seconds if absolutely necessary. Trials were stopped, excluded, and repeated if tapping occurred before 10 seconds or if the weight was fully shifted to the non-stance limb after 10 seconds. Individual tasks (stance limbs) were examined in a counterbalanced order within each group to ensure equal distribution and potential learning effect. Data from the first 10 seconds of each trial were used for all analyses.

#### 2.3.2. Movement Variability Measure

The SOT recorded COP coordinates in both the anteroposterior (AP) and mediolateral (ML) directions at 100 Hz for each set of three trials per SOT condition. Raw data were exported to Excel (version 360; Microsoft Corporation, Redmond, WA, USA) and then analyzed using a custom R program in RStudio (version 4.0.0; RStudio, Inc., Boston, MA, USA). Path length was calculated using equation (1), where *N* represents the number of data points, and *i* symbolizes each successive data point, based on Rhea et al. [13]. This involved summing the magnitude of the distance shift from the COP at each time point to the resultant vector created by the combined COP AP and ML over the first 10 seconds of the trials [13].

SampEN values were computed using the algorithm from Richman and Moorman [14] with parameters *m* = 3 and *r* = 0.2 for the COP path length time series. This metric was validated with a maximum relative error of less than 0.05, corresponding to 95% confidence intervals and representing a 10% SampEN estimate. SampEN values in human neurobiological systems typically range from 0 to 2. Lower SampEN values are associated with more predictable and periodic (rigid) neurobiological systems, whereas higher SampEN values indicate more unpredictable and random neurobiological systems [15,16,17].
(1)$$PathLength:\sum_{i=1}^{N=1}\sqrt{{\left({AP}_{i+1}-{AP}_{i}\right)}^{2}{+\left({ML}_{i+1}-{ML}_{i}\right)}^{2}}$$

### 2.4. Statistical Analysis

We performed a one-way analysis of variance (ANOVA) to assess group differences in demographic characteristics (age, height, body weight, physical activity level, the number of ankle sprains, and self-reported giving way) and Identification of Functional Ankle Instability (IdFAI). A 2 (group) × 3 (task) mixed-design repeated measures ANOVA was used to examine group differences in the neurobiological systems involved in postural control in double-, uninjured-, and injured-limb stances across the six SOT conditions. Pairwise comparisons were performed using Tukey post hoc analyses to explore interaction effects. For comparisons showing significant differences, Cohen’s *d* effect size (ES) along with 95% confidence intervals (CIs) were calculated to assess the magnitude of differences in SampEN values. The ES values were categorized as small (*d* ≤ 0.40), medium (*d* = 0.41–0.79), or large (*d* ≥ 0.80) [18]. All statistical analyses were conducted using SPSS software (version 27; IBM Corp, Armonk, NY, USA) with an a priori *α* level of 0.05. The normal distribution of the data was confirmed by the Kolmogorov–Smirnov test, which indicated normality for all variables (*p* > 0.05).

## 3. Results

No significant group differences were observed with respect to age, height, weight, or physical activity level (*p* > 0.05; Table 1).

There were significant group-by-task interactions for C1-V_n_S_f_P_f_ (F_2,80_ = 3.481, *p* = 0.036) and C5-V_a_S_f_P_m_ (F_2,80_ = 3.797, *p* = 0.027) but not for C2-V_a_S_f_P_f_, C3-V_d_S_m_P_f_, C4-V_d_S_f_P_m_, and C6-V_d_S_m_P_m_ (*p* > 0.05). Individuals with CAI had lower SampEN in the uninjured-limb stance (*p* = 0.002) for C1-V_n_S_f_P_f_ and in both the uninjured- (*p* < 0.001) and injured-limb (*p* = 0.031) stances for C5-V_a_S_f_P_m_ compared to healthy controls (Table 3).

Significant main effects of group (F_1,40_ range = 4.743–7.254, *p* range = 0.010–0.035) and task (F_2,80_ range = 115.517–736.981, *p* < 0.001) were found for C2-V_a_S_f_P_f_, C3-V_d_S_m_P_f_, C4-V_d_S_f_P_m_, and C6-V_d_S_m_P_m_. Individuals with CAI had significantly lower SampEN for C2-V_a_S_f_P_f_ (*p* = 0.012), C3-V_d_S_m_P_f_ (*p* = 0.035), C4-V_d_S_f_P_m_ (*p* = 0.010), and C6-V_d_S_m_P_m_ (*p* = 0.028) than healthy controls (Table 4). Regardless of group, significantly greater SampEN was found in the injured- and uninjured-limb stances for C2-V_a_S_f_P_f_ (injured: *p* < 0.001, uninjured: *p* < 0.001), C3-V_d_S_m_P_f_ (injured: *p* < 0.001, uninjured: *p* < 0.001), C4-V_d_S_f_P_m_ (injured: *p* < 0.001, uninjured: *p* < 0.001), and C6-V_d_S_m_P_m_ (injured: *p* < 0.001, uninjured: *p* < 0.001) compared to the double-limb stance (Table 5).

## 4. Discussion

The primary findings of the study revealed that individuals with CAI exhibited less complexity (decreased SampEN values) involved in postural control during specific SOT conditions compared to healthy controls as supported by our hypothesis. Specifically, this was observed in the uninjured-limb stance during C1-V_n_S_f_P_f_ with the normal vision-fixed surroundings-fixed platform and in both the uninjured- and injured-limb stances during C5-V_a_S_f_P_m_ with an absent vision-fixed surroundings-moving platform (Table 3). This decrease in SampEN indicates more predictable and periodic (rigid) neurobiological systems in individuals with CAI compared to healthy controls. For C2-V_a_S_f_P_f_ with the absent vision-fixed surroundings-fixed platform, C3-V_d_S_m_P_f_ with the distorted vision-moving surroundings-fixed platform, C4-V_d_S_f_P_m_ with the distorted vision-fixed surroundings-moving platform, and C6-V_d_S_m_P_m_ with the distorted vision-moving surroundings-moving platform, significant main effects were found for both group and task factors. Greater complexity was consistently found in the uninjured- and injured-limb stances compared to the double-limb stance, regardless of the group (Table 5). In addition, the CAI group displayed less complexity than healthy controls during C2-V_a_S_f_P_f_, C3-V_d_S_m_P_f_, C4-V_d_S_f_P_m_, and C6-V_d_S_m_P_m_, regardless of task constraints (Table 4). However, the weak effect sizes (Ess = 0.25–0.50) of the group main effect with even 95% CIs crossing zero suggest that these differences, while present, may not be robust or practically meaningful, indicating that further research is needed to confirm the group main effect. Overall, group differences in the neurobiological systems involved in postural control were influenced by task constraints primarily when all sensory feedback was intact during the uninjured-limb stance and when vestibular feedback became dominant during both uninjured- and injured-limb stances.

There are a limited number of studies that have examined postural control using nonlinear methods with entropy metrics (i.e., SampEN) in individuals with CAI. One study reported decreased SampEN values in resultant COP velocity (COPV) in the double-limb stance and mediolateral COPV and resultant COPV during the injured-limb stance compared to healthy controls [19]. Conversely, another research group reported no group differences in SampEN values in anteroposterior or mediolateral COPV during an injured-limb stance with eyes closed [20]. Our results provide support for both findings by showing that the CAI group had lower SampEN values of COP path length in the uninjured-limb stance during C1-V_n_S_f_P_f_, in both the uninjured- and injured-limb stances during C5-V_a_S_f_P_m_, and in all remaining conditions (i.e., C2-V_a_S_f_P_f_, C3-V_d_S_m_P_f_, C4-V_d_S_f_P_m_, and C6-V_d_S_m_P_m_), regardless of limb stance.

This study is part of a larger experiment, and our CAI participants did not display somatosensory deficits based on sensory reweighting (reliance) ratios computed during SOT. They maintained posture very similar to, or even better than, healthy controls (no postural deficits) in double-, uninjured-, and injured-limb stances [8]. Therefore, it was unexpected to find reduced complexity during C1-V_n_S_f_P_f_, where all sensory feedback was available to maintain posture, especially in the uninjured-limb stance. Our previous study demonstrated a compatible sensory reweighting system during the double-limb stance in both groups, whereas the CAI group failed to downweight vestibular feedback in the injured-limb stance [8]. A moderate trend of failure to downweight vestibular feedback was also found in the uninjured-limb stance among the CAI group [8].

Current evidence suggests that postural control becomes modulated by supraspinal mechanisms as tasks become more difficult, aiming for better precision to ensure dynamic stability at the ankle [2,3]. Although there were no group differences in balance scores in all limb stance types during C1-V_n_S_f_P_f_, reduced complexity in the uninjured-limb stance during the condition in which all sensory feedback was intact may indicate a centrally mediated change in multisensory integration that limits the number of configurations available to the CNS. Similarly, reduced complexity was observed in both uninjured- and injured-limb stances compared to healthy controls during C5-V_a_S_f_P_m_, where participants were forced to rely solely on vestibular feedback. This likely occurred because the vestibular feedback, which the CAI group relied on to maintain posture in the uninjured- and injured-limb stances, became the dominant sensory feedback available while performing C5-V_a_S_f_P_m_. As a result, the CAI group maintained better posture than the healthy controls [8].

Existing research suggests that long-term reductions in complexity may lead to abnormal configurations of the sensory cortex [21,22,23]. We cannot rule out the possibility that an abnormal sensory cortex configuration was present prior to sustaining an initial ankle sprain. Additionally, all our participants reported completing some form of rehabilitation with allied health professionals following ankle sprains. However, reliance on vestibular feedback in both the uninjured- and injured-limb stances may indicate a change in sensory feedback configuration [24,25,26,27,28,29]. Furthermore, the reduced complexity exhibited in the uninjured-limb stance, supported by strong effect sizes (Table 3) during C1-V_n_S_f_P_f_ (ES = 0.99) and C5-V_a_S_f_P_m_ (ES = 1.24), may be evidence of centrally mediated alterations after completion of rehabilitation in the injured limb. Indeed, postural control deficits have been reported not only in the injured-limb stance but also in the uninjured-limb stance in those with CAI [30].

A significantly reduced complexity was demonstrated in the double-limb stance compared to uninjured- and injured-limb stances, regardless of group during C2-V_a_S_f_P_f_, C3-V_d_S_m_P_f_, C4-V_d_S_f_P_m_, and C6-V_d_S_m_P_m_ (Table 5). This suggests that the CNS may integrate sensory feedback in a way that reduces complexity when both limbs are used simultaneously. Pasma et al. [31] found that proprioceptive feedback from the left and right limbs was independently weighted when the unilateral limb was perturbed while maintaining posture in a double-limb stance. Moreover, the study found that the down-weighting of proprioceptive information from one leg was accompanied by an upweighting of the vestibular feedback and not an upweighting of proprioceptive information from the contralateral leg [31]. We do not know whether somatosensory feedback is independently weighted for each limb when both limbs are perturbed simultaneously while maintaining posture in a double-limb stance; however, it is our hypothesis that the reduced complexity exhibited in double-limb stance may result from the CNS attempting to integrate sensory feedback unilaterally. There is redundant sensory feedback available from the proximal joints that is not limited to the contralateral limb; thus, there are infinite ways to integrate sensory information. Collectively, reduced complexity during a bipedal stance may indicate a reduction in sensory redundancy by limiting the number of available configurations, enabling the CNS to select the most reliable sensory information to maintain posture.

Based on our findings, it may be critical to incorporate rehabilitation strategies that target not only the injured limb but also the uninjured limb in individuals with CAI. Rehabilitation programs should include sensory manipulations with increased task complexity, particularly emphasizing vestibular feedback. Such interventions may help to improve the lower neurobiological systems complexity of postural control observed in the CAI group. By challenging the postural control system under varying sensory conditions, particularly in unilateral stances, clinicians may be able to improve the overall stability and adaptability of the neurobiological systems in those with CAI, thereby reducing the risk of future ankle sprains and enhancing functional outcomes.

One limitation of our study is that we tested physically active young adults with and without CAI. Another limitation was that we did not examine physiological aspects such as lower extremity muscle strength, limb dominance, or supraspinal mechanisms that may have contributed to the neurobiological systems. It is still not fully understood how primary sensory feedback (somatosensory, visual, and vestibular) is integrated by the CNS to maintain posture. Future studies should consider investigating the mechanisms of the sensory reweighting system that contribute to the neurobiological systems during postural control in individuals without CAI.

## 5. Conclusions

Our study revealed that specific task constraints influence the neurobiological systems involved in postural control. Individuals with CAI exhibited reduced complexity during the uninjured-limb stance when all sensory feedback was intact and during both uninjured- and injured-limb stances when they were forced to rely on vestibular feedback. These findings suggest that the neurobiological systems in individuals with CAI may undergo adaptive changes, possibly due to repeated ankle sprains, which could result in a more predictable and less adaptive postural control strategy when sensory feedback is manipulated. Future research should explore the mechanisms of the sensory reweighting system and the complexity of the neurobiological systems involved in postural control in individuals with and without CAI. A deeper understanding of these mechanisms could provide valuable insights for the development of more targeted and effective rehabilitation strategies using a multisensory feedback approach.

## Figures and Tables

**Table 1 bioengineering-11-00956-t001:** Demographics of participants and patient-reported outcome measures (mean ± SD).

	Group		
	Control	CAI	*p*-Values
N	21 (13 females, 8 males)	21 (13 females, 8 males)	-
Age (years)	25.41 ± 5.92	26.09 ± 5.76	0.836
Height (cm)	169.70 ± 9.32	172.25 ± 9.76	0.606
Weight (kg)	71.98 ± 14.79	76.18 ± 14.91	0.934
NASA-PASS	6.27 ± 1.03	6.27 ± 0.18	0.674
IdFAI	1.36 ± 1.81	19.09 ± 5.39	<0.001 *
Number of ankle sprains	0.00 ± 0.00	6.48 ± 7.08	<0.001 *
Episodes of giving way	0.00 ± 0.00	8.88 ± 21.36	<0.001 *

Abbreviations: CAI, Chronic Ankle Instability; NASA-PASS, National Aeronautics and Space Administration Physical Activity Status Scale; IdFAI, Identification of Functional Ankle Instability. * Indicates a significant difference between the CAI group and healthy controls.

**Table 2 bioengineering-11-00956-t002:** Descriptions of six sensory organization test conditions.

SOT Conditions	Sensory Feedback					
	Manipulation Modalities			Manipulated	Absent	Tested
	Support Surface	Eyes	Visual Surroundings			
C1-V_n_S_f_P_f_	Fixed	Open	Fixed	None	-	None
C2-V_a_S_f_P_f_	Fixed	Closed	Fixed	None	VIS	SOM
C3-V_d_S_m_P_f_	Fixed	Open	Sway-referenced	VIS	-	SOM
C4-V_d_S_f_P_m_	Sway-referenced	Open	Fixed	SOM	-	VIS
C5-V_a_S_f_P_m_	Sway-referenced	Closed	Fixed	SOM	VIS	VST
C6-V_d_S_m_P_m_	Sway-referenced	Open	Sway-referenced	SOM, VIS	-	VST

Abbreviations: SOT, Sensory Organization Test; C1-V_n_S_f_P_f_, Condition 1-Normal Vision-Fixed Surroundings-Fixed Platform; C2-V_a_S_f_P_f_, Condition 2-Absent Vision-Fixed Surroundings-Fixed Platform; C3-V_d_S_m_P_f_, Condition 3-Distorted Vison-Moving Surroundings-Fixed Platform; C4-V_d_S_f_P_m_, 4-Distorted Vision-Fixed Surroundings-Moving Platform; C5-V_a_S_f_P_m_, Condition 5-Absent Vision-Fixed Surroundings-Moving Platform; C6-V_d_S_m_P_m_, Condition 6-Distorted Vision-Moving Surroundings-Moving Platform; VIS, Vision; SOM, Somatosensory; VST, Vestibular.

**Table 3 bioengineering-11-00956-t003:** A two-factor interaction for group and task constraints: pairwise comparisons of neurobiological systems complexity between groups.

SampEN Values					
Parameter		Group		*p*-Value	Effect Size (95% CI)
SOT Condition	Task	Control	CAI		
C1-V_n_S_f_P_f_	Double-limb	1.04 ± 0.13	0.98 ± 0.13	0.213	0.46 (−0.14 to 1.06)
	Uninjured-limb	1.34 ± 0.10	1.22 ± 0.14	0.002 *	0.99 (0.36 to 1.61)
	Injured-limb	1.30 ± 0.12	1.23 ± 0.13	0.084	0.56 (−0.04 to 1.16)
C5-V_a_S_f_P_m_	Double-limb	1.38 ± 0.12	1.32 ± 0.10	0.076	0.54 (−0.06 to 1.15)
	Uninjured-limb	1.91 ± 0.15	1.73 ± 0.14	<0.001 *	1.24 (0.60 to 1.89)
	Injured-limb	1.85 ± 0.13	1.75 ± 0.17	0.031 *	0.66 (0.05 to 1.27)

Abbreviations: SampEN, Sample Entropy; SOT, Sensory Organization Test; C1-V_n_S_f_P_f_, Condition 1-Normal Vision-Fixed Surroundings-Fixed Platform; C5-V_a_S_f_P_m_, Condition 5-Absent Vision-Fixed Surroundings-Moving Platform; CAI, Chronic Ankle Instability; CI, Confidence Interval. * Indicates a significant difference.

**Table 4 bioengineering-11-00956-t004:** Group main effect: pairwise comparisons of neurobiological systems complexity between groups.

SampEN Value				
Parameter	Group		*p*-Value	Effect Size (95% CI)
SOT Condition	Control	CAI		
C2-V_a_S_f_P_f_	1.52 ± 0.33	1.44 ± 0.32	0.012 *	0.25 (−0.35 to 0.84)
C3-V_d_S_m_P_f_	1.37 ± 0.25	1.29 ± 0.24	0.035 *	0.33 (−0.27 to 0.92)
C4-V_d_S_f_P_m_	1.30 ± 0.18	1.21 ± 0.18	0.010 *	0.50 (−0.10 to 1.10)
C6-V_d_S_m_P_m_	1.50 ± 0.20	1.42 ± 0.19	0.028 *	0.30 (−0.19 to 1.01)

Abbreviations: SampEN, Sample Entropy; SOT, Sensory Organization Test; C2-V_a_S_f_P_f_, Condition 2-Absent Vision-Fixed Surroundings-Fixed Platform; C3-V_d_S_m_P_f_, Condition 3-Distorted Vison-Moving Surroundings-Fixed Platform; C4-V_d_S_f_P_m_, 4-Distorted Vision-Fixed Surroundings-Moving Platform; C6-V_d_S_m_P_m_, Condition 6-Distorted Vision-Moving Surroundings-Moving Platform; CAI, Chronic Ankle Instability; CI, Confidence Interval. * Indicates a significant difference.

**Table 5 bioengineering-11-00956-t005:** Task main effect: pairwise comparisons of neurobiological systems complexity between task constraints.

SampEN Value					
Parameter					
SOT Condition	Task			*p*-Value	Effect Size (95% CI)
	Double	Uninjured	Injured		
C2-V_a_S_f_P_f_	1.05 ± 0.12	1.69 ± 0.14	-	<0.001 *	4.91 (3.72 to 6.09)
C2-V_a_S_f_P_f_	1.05 ± 0.12	-	1.69 ± 0.13	<0.001 *	5.12 (3.89 to 6.34)
C2-V_a_S_f_P_f_	-	1.69 ± 0.14	1.69 ± 0.13	0.569	0.00 (−0.59 to 0.59)
C3-V_d_S_m_P_f_	1.04 ± 0.14	1.49 ± 0.15	-	<0.001 *	3.10 (2.22 to 3.98)
C3-V_d_S_m_P_f_	1.04 ± 0.14	-	1.46 ± 0.14	<0.001 *	3.00 (2.14 to 3.86)
C3-V_d_S_m_P_f_	-	1.49 ± 0.15	1.46 ± 0.14	0.063	0.21 (−0.39 to 0.80)
C4-V_d_S_f_P_m_	1.09 ± 0.14	1.35 ± 0.14	-	<0.001 *	1.86 (1.15 to 2.56)
C4-V_d_S_f_P_m_	1.09 ± 0.14	-	1.34 ± 0.14	<0.001 *	1.79 (1.09 to 2.48)
C4-V_d_S_f_P_m_	-	1.35 ± 0.14	1.34 ± 0.14	0.427	0.07 (−0.52 to 0.66)
C6-V_d_S_m_P_m_	1.26 ± 0.11	1.57 ± 0.17	-	<0.001 *	2.17 (1.42 to 2.91)
C6-V_d_S_m_P_m_	1.26 ± 0.11	-	1.55 ± 0.14	<0.001 *	2.30 (1.54 to 3.07)
C6-V_d_S_m_P_m_	-	1.57 ± 0.17	1.55 ± 0.14	0.230	0.13 (−0.46 to 0.72)

Abbreviations: SampEN, Sample Entropy; SOT, Sensory Organization Test; C2-V_a_S_f_P_f_, Condition 2-Absent Vision-Fixed Surroundings-Fixed Platform; C3-V_d_S_m_P_f_, Condition 3-Distorted Vison-Moving Surroundings-Fixed Platform; C4-V_d_S_f_P_m_, 4-Distorted Vision-Fixed Surroundings-Moving Platform; C6-V_d_S_m_P_m_, Condition 6-Distorted Vision-Moving Surroundings-Moving Platform; CAI, Chronic Ankle Instability; CI, Confidence Interval. * Indicates a significant difference.

## Data Availability

Data are included in the article and are available upon reasonable request.

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
