# Peer review of "Effect of Task Constraints on Neurobiological Systems Involved in Postural Control in Individuals with and without Chronic Ankle Instability"

_bioengineering, 2024, doi:10.3390/bioengineering11100956_

Round 1
Reviewer 1 Report
Comments and Suggestions for Authors
The purpose of the study was to investigate the impact of task constraints on the neurobiological system while maintaining postural control under various environmental constraints, employing sensory feedback manipulations among individuals with and without Chronic Ankle Instability (CAI).
The article is too confusing. The title should be specified better. The title is unclear. The design of the study is not well understood and the introduction and conclusions should also be better articulated
Individuals with Chronic Ankle Instability (CAI) commonly exhibit postural control (stability, adaptation) deficits and altered gait (walking, running) mechanics. Moreover, it may optimize rehabilitation intervention to prevent subsequent ankle sprains in individuals with CA. The results highlight the importance of considering both environmental and task constraints when evaluating postural control deficits in individuals with CAI, but reviewed in the future and research but do not indicate which path may be significant.
The methodology is detailed but it is necessary to follow a methodological logic; It is performed is too complicated to implement
Author Response
Comments 1: The purpose of the study was to investigate the impact of task constraints on the neurobiological system while maintaining postural control under various environmental constraints, employing sensory feedback manipulations among individuals with and without Chronic Ankle Instability (CAI).
Response 1: Thank you for recognizing the study’s purpose. To clarify, our research aimed to investigate the effect of task constraints on the neurobiological system while maintaining postural control under various environmental constraints using sensory feedback manipulations in individuals with and without CAI.
Comments 2: The article is too confusing. The title should be specified better. The title is unclear. The design of the study is not well understood and the introduction and conclusions should also be better articulated.
Response 2: We appreciate your feedback regarding the clarity of the title, study design, introduction, and conclusions. We have revised the title to better reflect the focus of the study: "The Effect of Task Constraints on Neurological System Involved in Postural Control in Individuals with Chronic Ankle Instability." This revised title more accurately reflects the focus of our study. We have also updated the study design section. The introduction has been revised to better articulate the background, rationale, and significance of the study. Additionally, we have clarified the conclusions to explicitly highlight the key findings and future research.
Comments 3: Individuals with Chronic Ankle Instability (CAI) commonly exhibit postural control (stability, adaptation) deficits and altered gait (walking, running) mechanics. Moreover, it may optimize rehabilitation intervention to prevent subsequent ankle sprains in individuals with CA. The results highlight the importance of considering both environmental and task constraints when evaluating postural control deficits in individuals with CAI, but reviewed in the future and research but do not indicate which path may be significant.
Response 3: To address your concerns, we have updated the discussion section to provide more specific recommendations for rehabilitation strategies and highlight potential research paths that could further explore the implications of our results.
Comments 4: The methodology is detailed but it is necessary to follow a methodological logic; It is performed is too complicated to implement.
Response 4: We appreciate your feedback regarding the methodology. We have restructured the methodology section to present a clearer and more logical sequence of steps, ensuring that each aspect of our approach is well-justified and easy to follow. We acknowledge that implementing the Sensory Organization Test (SOT) and Sample Entropy analysis may appear complex. However, these methods are well-established in the literature. The SOT, introduced by Nashner in 1985, has been extensively utilized in similar research (e.g., Sugimoto et al., 2024; Yin and Want, 2020; Song et al., 2020). Similarly, Sample Entropy has been validated in previous studies (e.g., Rhea et al., 2014; Richman and Moorman, 2000). We have referenced these works to demonstrate the robustness and reliability of our methodological approach, emphasizing its relevance and feasibility.

Reviewer 2 Report
Comments and Suggestions for Authors
The hypotheses of the study are not framed and clearly reported at the end of introduction; therefore, the study is not based on previous hypotheses framed on a scientific rationale and previous findings.
IRB number and date has to be reported.
Which is the rationale to allow participants “to tap down with non-stance toes after 10 seconds” in a 20-second test. Could this instruction influence the entire execution of test and the overall evaluation?
Trials have not been randomized. Considering the total number of 18 trials per stance, could the lack of randomization have influenced the evaluation?
Effect size should be declared in statistical analysis and further interpreted in discussion.
The author did not provided tangible practical application from the current study
Reference list is not written following the journal format and guidelines.
Comments on the Quality of English LanguageQuality of scientific and academic writing style is sufficient.
Author Response
Comments 1: The hypotheses of the study are not framed and clearly reported at the end of introduction; therefore, the study is not based on previous hypotheses framed on a scientific rationale and previous findings.
Response 1: We appreciate your feedback. To address this, we have revised the introduction to clearly state our hypotheses at the end of the section. Our study hypothesized that that individuals with CAI will demonstrate significantly lower complexity in the neurological system as task constraints increase, particularly under greater sensory feedback manipulations, while maintaining posture in the injured-limb stance compared to double- and uninjured- limb stances.
Comments 2: IRB number and date has to be reported.
Response 2: The study was approved by IRB under the number 19-0446, with an approval date of 2/25/19. We prefer not to include that information in the manuscript unless it is absolutely required by the journal. Although we generally prefer not to include IRB details unless explicitly required by the journal, we are fully willing to incorporate this information if it is mandated by the journal.
Comments 3: Which is the rationale to allow participants “to tap down with non-stance toes after 10 seconds” in a 20-second test. Could this instruction influence the entire execution of test and the overall evaluation?
Response 3: The Sensory Organization Test (SOT) is traditionally designed for double-limb stances, making it particularly challenging to maintain posture on a unilateral stance with a sway-referenced support surface, especially for individuals with CAI. Following established protocols (Hu et al., 2023; Ekstrom et al., 2017; Lines et al., 2014), we permitted a brief tap-down after 10 seconds to prevent undue fatigue and capture their true postural control abilities over the most critical period if it was necessary. Participants were encouraged to complete the full 20-second trials. Importantly, our data analysis focused solely on the first 10 seconds of each trial, ensuring that any tap-down did not influence the overall test execution or its evaluation. This protocol is consistent with established practices and allows for a robust assessment of postural control in individuals with CAI.
Comments 4: Trials have not been randomized. Considering the total number of 18 trials per stance, could the lack of randomization have influenced the evaluation?
Response 4: We acknowledge the concern regarding trial randomization. The SOT system does not permit randomization of the trials themselves. However, we mitigated potential biases by counterbalancing the order of stance limbs within each group. This approach was designed to minimize any learning effects or biases, thereby ensuring that the difficulty of tasks was consistently distributed across participants.
Comments 5: Effect size should be declared in statistical analysis and further interpreted in discussion.
Response 5: We have revised the statistical analysis section to include effect size estimates for our primary outcomes. In the discussion, we have focused on interpreting the effect sizes that were both meaningful and significant.
Comments 6: The author did not provided tangible practical application from the current study
Response 6: In response to your suggestion, we have revised the discussion section to include tangible and concrete practical applications derived from our findings. These applications emphasize the importance of considering uninjured limb rehabilitation and incorporating sensory manipulations with increased task complexity, particularly focusing on vestibular feedback during postural control tasks.
Comments 7: Reference list is not written following the journal format and guidelines.
Response 7: We have revised the reference list to adhere to the journal’s format and guidelines.

Round 2
Reviewer 1 Report
Comments and Suggestions for Authors
The purpose of the study was to investigate the impact of task constraints on the neurobiological system while maintaining postural control under various environmental constraints, employing sensory feedback manipulations among individuals with and without Chronic Ankle Instability (CAI). After revision the manuscript can be accept in present form
Reviewer 2 Report
Comments and Suggestions for Authors
Hypothesis should be framed before the start of any research.
I cannot accept that the authors framed the hypothesis after the submission of the manuscript. This is a critical flaw for me and the reason of the first rejection.
I leave the editor the decision to accept that the authors did not frame the hypothesis before the first submission.
Comments on the Quality of English LanguageQuality of scientific and academic writing style is sufficient.